# The contribution of object identity and configuration to scene representation in convolutional neural networks

**Kevin Tang, Matthew Chin, Marvin Chun, Yaoda Xu** *

Department of Psychology, Yale University, New Haven, CT, United States of America

* yaoda.xu@yale.edu

**Data Availability Statement:** All materials and data supporting the findings reported in this study are available at https://osf.io/7q5pj/.

## Abstract

Scene perception involves extracting the identities of the objects comprising a scene in conjunction with their configuration (the spatial layout of the objects in the scene). How object identity and configuration information is weighted during scene processing and how this weighting evolves over the course of scene processing however, is not fully understood. Recent developments in convolutional neural networks (CNNs) have demonstrated their aptitude at scene processing tasks and identified correlations between processing in CNNs and in the human brain. Here we examined four CNN architectures (Alexnet, Resnet18, Resnet50, Densenet161) and their sensitivity to changes in object and configuration information over the course of scene processing. Despite differences among the four CNN architectures, across all CNNs, we observed a common pattern in the CNN's response to object identity and configuration changes. Each CNN demonstrated greater sensitivity to configuration changes in early stages of processing and stronger sensitivity to object identity changes in later stages. This pattern persists regardless of the spatial structure present in the image background, the accuracy of the CNN in classifying the scene, and even the task used to train the CNN. Importantly, CNNs' sensitivity to a configuration change is not the same as their sensitivity to any type of position change, such as that induced by a uniform translation of the objects without a configuration change. These results provide one of the first documentations of how object identity and configuration information are weighted in CNNs during scene processing.

## Introduction

A fundamental task for any intelligent system is to interact with its environment in an adaptive manner. To accomplish this, the system needs to understand and extract useful information from the environment in order to guide its interaction with the environment. One such system, the human brain does a remarkable job of extracting and filtering the relevant information out of a scene, enabling us to understand and navigate the world around us. Besides the brain regions involved in general visual information processing, research from the past several decades has identified three key regions in the human brain, the parahippocampal place area

**Funding:** The project is supported by NIH grants 1R01MH108591 to MC and 1R01EY030854 to YX. The funders had no role in study design, data collection and analysis, decision to publish, or preparation of the manuscript.

**Competing interests:** The authors have declared that no competing interests exist.

(PPA) [1], the retrosplenial complex/medial place area (RSC/MPA) [2], and the occipital place area (OPA) [3], as comprising the human scene network due to their strong responses during scene processing [4, 5]. While all three brain regions are involved in the processing of spatial layouts within a scene, the PPA and OPA are activated during analysis of a scene's local spatial structure whereas RSC/MPA is concerned with understanding the relationship between local scene and the broader environment [4, 6; see also [5] for a new proposal regarding the distinct roles of these regions).

Recent developments in convolutional neural networks (CNNs) have demonstrated that artificial visual processing systems are capable of performing scene and object recognition tasks with high accuracy [7, 8]. Thus, the ability to extract the relevant information for accurate object and scene recognition is not limited to an organic brain, but can be accomplished in *silico*. Further research shows that mid to high-level features of a CNN trained to recognize objects can explain cortical responses in scene-selective areas in the human brain [9] (see also [10–12]). This suggests that despite hardware differences, similar computational algorithms may govern scene processing in both the human brain and CNNs. This provides us with an exciting opportunity to potentially understand the neural computational algorithms supporting human scene processing by studying the algorithms used by CNNs for scene processing.

In this study, we aim to understand the degree to which object identity or configuration information may dominate scene representation. A scene is often defined by the objects it contains and by the spatial configuration or layout of these objects in space, making both object identity and configuration information important components of a scene representation. Representing the objects' configuration in a scene, besides being important for scene recognition, is also critical for scene navigation and interaction. But to our knowledge, how object identity and configuration information are weighted during scene processing has not been systematically examined in either the human brain or CNNs. Would CNNs show sensitivity to configuration changes in scenes? On one hand, because CNNs are trained for scene classification and because rearranging the same set of objects into a new and reasonable configuration does not change the nature of the scene (e.g., rearranging the furniture in a bedroom will still yield a bedroom), CNNs may not encode the precise object arrangement in a scene and thus would demonstrate a lack of sensitivity to such a configuration change. On the other hand, if objects and configurations are fundamental elements of scene representation, then a system capable of classifying scenes should nevertheless explicitly encode these elements and demonstrate sensitivity to changes in both. Feedforward CNNs trained for scene classification thus provide us with an excellent opportunity to test this idea.

Motivated by the finding that scene processing in CNNs resembles that in the human brain [9], in this study, we investigated how a CNN responds to changes in object identity and configuration information within a scene. We documented both a CNN's absolute sensitivity to these changes using Euclidean distance measures and its relative sensitivity to these changes using an index measure adapted from [13]. We examined a total of four CNN architectures varying in depth, complexity, and degree of interconnectivity. They were Alexnet, Resnet18, Resnet50 and Densenet161. Alexnet was the CNN tested in [9] and its scene representations at the mid and high levels have been shown to resemble those of the scene processing regions in the human brain; this CNN was also examined in [10, 11] in their brain and CNN studies. [11] additionally examined Resnet18 and Resnet50 and found that these networks also exhibited similarities to processing found in human scene processing regions.

Taking full advantage of the accessibility of CNNs, we additionally investigated how a number of other factors may impact a CNN's sensitivity to object identity and configuration information. Specifically, to understand the effect of a spatially structured scene background, we compared responses from images with an indoor room background to images without such a

background. To understand how a CNN's ability to correctly classify a scene image could impact its scene representation, we compared responses from images eliciting high classification accuracies to those eliciting poor classification accuracies. To understand the effect of training, we compared responses from CNNs trained on scene recognition tasks to those from CNNs trained on object recognition tasks. We also compared responses from the scene trained CNNs to untrained (randomly weighted) CNNs. Lastly, to understand whether our results were specific to configuration changes or would arise from any type of object position changes, we compared our results with those from a uniform left or right spatial translation that preserved the configuration among the objects.

While CNNs are fully image computable and accessible, they are at the same time extremely complex models with millions or even hundreds of millions of free parameters. And while past research has focused primarily on image classification performance and the similarity between CNNs and the brain (e.g., [14–16]), the general operating principles at the algorithmic level [17] that enable CNNs' to succeed in object and scene categorization remain poorly understood (e.g., [18]). Documenting how object identity and configuration information is weighted during scene processing and how this weighting evolves over the course of CNN scene processing would thus bring us one step closer towards understanding CNN scene processing at the algorithmic level. This in turn may help unveil a set of general principles governing scene processing and allow us to then test whether similar algorithms may be implemented in the human brain during scene processing.

## Methods

### CNN selection

In this study, we examined the following CNNs: Alexnet, Resnet18, Resnet50, and Densenet161, where each was chosen due to its strong performance against both scene and object recognition tasks. In particular, Alexnet [19] was chosen for its simplicity and relatively shallow architecture. Resnet18 and Resnet50 [20] were chosen for their convolution depth and novel approach using residual blocks. Residual networks trained for object classification are considered to be the first to surpass human performance, with residual blocks helping the gradient flow by allowing it to bypass processing layers that would otherwise cause it to dissipate and disrupt learning and thereby enabling the model to learn appropriate processing depth for a task [14]. Densenet161 [21] was chosen for its complexity and high degree of interconnectivity, far surpassing the interconnectivity of even the residual blocks. Together, these networks provided a diversity of architectures with varying depth, complexity, and degree of interconnectivity. Previous brain and CNN studies have also included some of these networks, with Alexnet being examined in [9–11], and Resnet18 and Resnet50 being examined in [11]. For our study, we used the PyTorch implementation of each CNN architecture with pretrained-weights from both scene-recognition and object-recognition tasks. The weights from training on the scene-recognition task were obtained from the Places365 project [7]. The weights from training on the object-recognition task were taken from the PyTorch model library and resulted from training over the ImageNet Database [22].

Following previously published studies (e.g., [13, 16, 23–26]), we sampled six to seven CNN layers from various stages of the convolution. It has been shown previously that such a sampling procedure captures the evolution of the entire representation trajectory fairly well, if not fully, as adjacent layers exhibit identical or very similar representations [27]. When possible, layers were taken at natural transitions such as between blocks of highly interconnected processing. This included pooling layers, where unit responses are aggregated and downsampled. For all CNNs we included the classifier as the last sampled layer. In many cases, the

penultimate layer was also sampled. Table 1 lists the specific layers sampled for each model. Because the exact identity of each layer was not important to our analysis, each layer is referred to by its ordinal position inside the network hierarchy.

In addition to both scene and object trained variants of each CNN architecture, we also examined responses from an untrained variant of each architecture. As a single instance of randomly generated weights could yield results unrepresentative of the population, we averaged the results from 100 instances of each CNN initialized with randomly generated weights per the default PyTorch initialization algorithms. The result of this procedure was reported as the untrained variant of each CNN architecture.

## Stimulus construction

**Identity vs configuration manipulation.** We used indoor room scenes in the present study because these scenes are model examples of real-world scenes with easily manipulable object identity/configuration information, as the furniture in a room can be rearranged or swapped without changing the classification nature of the scene. Additionally, CNNs have been shown to classify different room scenes with high accuracy.

We constructed several collections of computer-generated images, with each containing 15 to 20 different image sets. Each image set consisted of four images, with each image depicted a

**Table 1. CNN architectures and their sampled layers.**

| CNN (total layers) | Position | Type | Layer Id (from PyTorch) |
|---|---|---|---|
| Alexnet (20) | 1 | MaxPool2d | features[2] |
| | 2 | MaxPool2d | features[5] |
| | 3 | MaxPool2d | features[12] |
| | 4 | Linear | classifier[1] |
| | 5 | Linear | classifier[4] |
| | 6 | Linear | classifier[6] |
| Resnet18 (52) | 1 | MaxPool2d | maxpool |
| | 2 | BatchNorm2d | layer1[1].bn2 |
| | 3 | BatchNorm2d | layer2[1].bn2 |
| | 4 | BatchNorm2d | layer3[1].bn2 |
| | 5 | BatchNorm2d | layer4[1].bn2 |
| | 6 | AvgPool2d | avgpool |
| | 7 | Linear | fc |
| Resnet50 (126) | 1 | MaxPool2d | maxpool |
| | 2 | Bottleneck | layer1[2] |
| | 3 | Bottleneck | layer2[3] |
| | 4 | Bottleneck | layer3[5] |
| | 5 | Bottleneck | layer4[2] |
| | 6 | AvgPool2d | avgpool |
| | 7 | Linear | fc |
| Densenet161 (486) | 1 | MaxPool2d | features[3] |
| | 2 | AvgPool2d | features[5].pool |
| | 3 | AvgPool2d | features[7].pool |
| | 4 | AvgPool2d | features[9].pool |
| | 5 | Conv2d | features[10].denselayer24.conv2 |
| | 6 | BatchNorm2d | features[11] |
| | 7 | Linear | classifier |

room scene with four pieces of furniture situated around the space–the arrangement of furniture around the space is referred to as the configuration. An image set was constructed from two sets of objects and two configurations. The two sets of objects were drawn from the same four furniture categories, with different exemplars appearing in each set. For example, if the first set of objects contained a lamp, a chair, a tv plus stand, and a rug, then the second set of objects would contain a different lamp, different chair, different tv plus stand, and different rug (see examples shown in Fig 1A and 1B). We crossed the two sets of objects with two unique configurations to produce the four room scenes for a given image set. Each set thus contained two pairs of images with the same objects in different configurations and two pairs of images with different objects but in the same configuration.

Following the procedure as outlined, we constructed four different collections of image sets. In the "fixed background" collection of 15 image sets, all images share a fixed indoor room background consisting of two visible gray walls and a birch floor (Fig 1A and 1B). These images were generated at 1883 x 1690 pixel resolution. To examine how the presence of spatial information in the background may impact identity and configuration representation, we removed the scene background from each image of this "fixed background" collection to create the "no background" collection (Fig 1C).

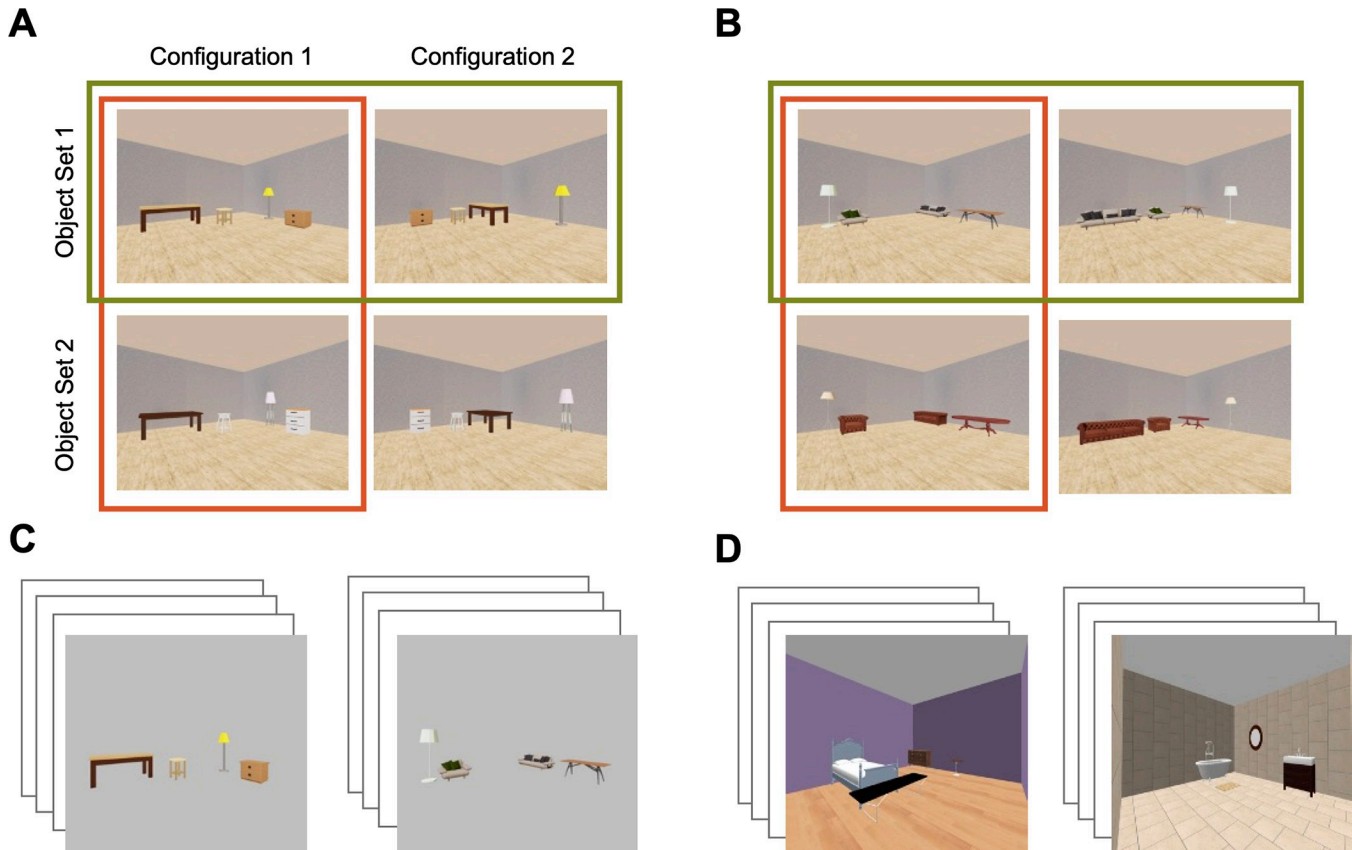

**Fig 1. Image collections with different background manipulations.** (**A**) and (**B**) Two example image sets from the "Fixed Background" image collection. All image sets in this collection had an identical room background. Each image set consisted of four unique images, constructed from two sets of objects and two configurations. Two images within a set could express the same configuration but with different objects (see the two images in the red box), or have the same objects but different configurations (see the two images in the green box). (**C**) Two example images from the "No Background" image collection. All image sets in this collection were shown without a room background. (**D**) Two example images from the "Variable Background" image collection. Each image set in this collection had a unique room background.

To understand how CNN classification accuracy may impact its identity and configuration representation, we also examined responses from a more realistic looking scene collection, termed the "variable background" collection which consisted of 20 distinct image sets constructed using the same procedure as described above (Fig 1D). These images were generated previously for a different study, and exhibit varying backgrounds across the different image sets. These variations took the form of color, texture, and material changes to the floor/walls/ceiling. These image sets were generated using different object sets from the preceding two collections, and were generated at a resolution of 600 x 800 pixel resolution. Limitations with the generation procedure also introduced small, unintended shifts in perspective across the four images in these sets. These differences prevented us from directly comparing results from the "variable background" collection to the results obtained from our first two image collections (however these differences did not prevent us from replicating the general effects found with the first two image sets as described in the Results).

**Configuration vs translation manipulation.** To understand whether our results were uniquely associated with a configuration change, which is a relative position change, we compared configuration changes to uniform spatial translations. "Uniform translations" shift the absolute position of all objects in a scene by the same amount, representing a class of position changes which preserve configuration.

To examine the differences between configuration and translation, we generated the "spatial shift" collection. Given similar response profiles between the "fixed background" and "no background" image collections (see Results), to facilitate image construction, we constructed images without a spatially structured scene background by using the image sets of the "no background" collection. Additionally, to produce larger and more distinguishable translations, we reduced the object sizes by 40%. Comparing the CNN object dominance indices of the reduced size image sets to the originals with a repeated measure ANOVA, we found no main effect of object size ($Fs < .61$, $ps > .43$) in any CNN nor an interaction between object size and layer ($Fs < 1.26$, $ps > .28$) in Alexnet, Resnet18, and Resnet50. There was an interaction effect of object size and layer in Densenet161 ($F = 2.32$, $p = .035$), which was due to a small crossover of response magnitude in the index measures for the early and later layers between the original and the size-reduced image sets. Because the effect is relatively small and the overall response profile was preserved across the size change, we felt justified to use the size-reduced images from the "no background" collection to construct the images for the "spatial shift" collection.

To construct the images for the "spatial shift" collection, we took the size-reduced images from the "no background" collection and uniformly translated all objects either to the left or the right. This resulted in a new set of eight images in each image set that varied in object identity, configuration and translation (left or right) (see Fig 2B), enabling us to directly compare the effect of configuration changes to translations.

The fixed background, no background, and spatial shifted image collections were generated using Unreal Engine version 4.23 developed by Epic Games. We built the walls, floor, and ceiling of the room using static mesh objects with different materials. For example, rectangular wood objects were used to comprise the floor. All images shown in these collections were taken from the exact same perspective using the same lighting to showcase the 3D structure of the room and the furniture it contains. All furniture was hand-picked from Unreal Engine's free object library to best represent typical objects in various indoor room scenes, e.g. couches in the living room. The variable background image collection was created using "us.mydeco.com" in 2011. This website appears to be no longer accessible. The full set of scene images are included in the data depository.

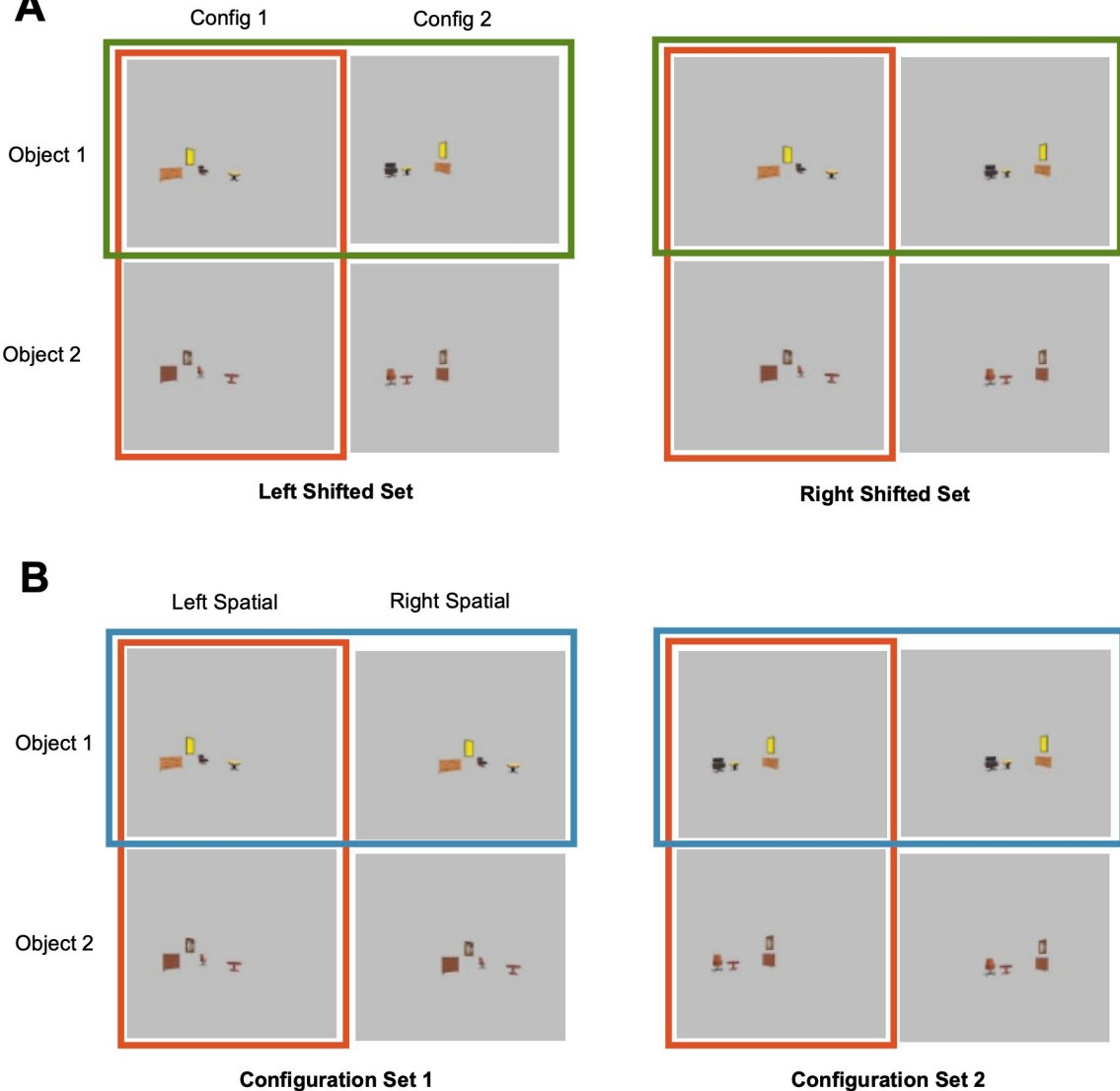

**Fig 2. An example image set showing object configuration change and spatial shift within the same set of images.** These images were modified from the "No Background" image shown in Fig 1. Each image set contained eight different images varying in configuration, objects, and spatial shift. (**A**) The grouping of the images based on the direction of the spatial shift (left or right). In Left Shifted Set, all four images had the same amount of leftward spatial shift but varied in object identity and configuration (e.g., the two images in the red box varied in object identity and the two in the green box varied in configuration). The same applied to the images in Right Shifted Set. (**B**) The grouping of the images based on configuration. In Configuration Set 1, all four images had the same configuration but varied in object identity and spatial shift (e.g., the two images in the red box varied in object identity and the two in the blue box varied in spatial shift). The same applied to the images in Configuration Set 2.

## Computing CNN classification accuracy

To understand how a CNN's classification accuracy for the scene images we used may impact its representation for these images, we calculated a top-5 scene classification accuracy of each scene-trained CNN for each image collection (see Table 2). We obtained each CNN's top five classification labels for a given image and assessed whether any of these labels would represent a reasonable classification of the scene. For example, in Fig 1A, the reasonable classifications of the image set were taken to be "dining room" and "living room" due to the presence of a large

**Table 2. Top 5 accuracy of each CNN.**

| CNN | Fixed Background | No Background | Variable Background |
|---|---|---|---|
| Alexnet | 0.400<br>0.200 | 0.267 | 0.550 |
| Resnet18 | 0.417<br>0.383 | 0.033 | 0.388 |
| Resnet50 | 0.483<br>0.567 | 0.083 | 0.488 |
| Densenet161 | 0.567<br>0.600 | 0.100 | 0.563 |

Values are expressed as a number between 0 and 1, indicating the proportion of successful classifications to total attempts (successes and failures) for a given scene collection (see Methods for details). The first row indicates the classification accuracy for the scene trained variant of each CNN; for the fixed background collection, the classification accuracy for the object trained variant of each CNN was reported in the second row. Whereas scene classification induces the CNN to supply scene labels taken from the Places365 image set, object classification induces the CNN to supply object labels from the ImageNet database.

table and seating. If a reasonable classification was included among the CNN's top-5 classifications, we recorded a classification success, and if not, a failure. This was done separately for each of the four images in a set and the results were averaged over all images and image sets to generate an average classification accuracy for each scene-trained CNN over a given image collection.

We also calculate an object classification accuracy for the object trained CNNs. As with scene classification, we obtained each CNN's top five classification labels for a given image and assessed whether any of these labels would represent a reasonable classification of any object present in a scene. For example, in Fig 1A, the reasonable object classifications were taken to be "wardrobe", "dining table", "desk", "lampshade", and "filing cabinet". If the object trained CNN identified any of these labels in its top-5 classifications, we considered the classification to be a success, and if not, a failure. This was done separately for each of the four images in a set and the results were averaged over all images and image sets to generate an average classification accuracy for each object-trained CNN over the fixed background image collection.

## Analysis

We conducted our analysis with the PyTorch implementations of our selected CNN architectures (Alexnet, Resnet18, Resnet50, Densenet161). Before feeding images to each network, we stretched and resized the dimensions of each image to 224 x 224 pixels, then normalized each as a tensor (mean = [0.485, 0.456, 0.406], std = [0.229, 0.224, 0.225]).

To obtain CNN response patterns for each image across different stages of processing, we first extracted CNN unit responses for each image from each sampled layer of the CNN. We then z-normalized these unit responses to remove amplitude differences across images, layers, architectures to obtain the CNN response pattern for a given image at a given layer. Within each image set, we measured the CNN's sensitivity to object identity and configuration changes through two Euclidean distance measures. Specifically, for each sampled layer, we calculated (1) the Euclidean distance of unit responses for images sharing the same configuration but different objects, and (2) the Euclidean distance of unit responses for images sharing the same objects but different configurations. A value of zero for measure (1) would indicate that the internal representation of a CNN does not change in response to a change in object identity, and hence the CNN is insensitive to this type of object identity change. For a given object identity change, a greater magnitude for measure (1) would indicate a stronger response to the

object identity change, hence a greater sensitivity. The same logic applies to measure (2) for characterizing configuration changes.

To facilitate between layer comparisons, these Euclidean distance measures were corrected for the number of units in each layer. This was done by dividing each distance measure by 2 times the square root of the total number of units in that layer–this was necessary as the distance between two opposite patterns would increase with the increasing number of units [13].

To quantify the relative contribution of object identity and configuration information at a given layer independent of the absolute magnitudes of the different Euclidean distance measures, following [13], we defined an *object dominance index* as [(1)—(2)] / [(1) + (2)]. If a CNN layer responds only to the object identity changes and is completely insensitive to configuration changes, then measure (2) would be zero and the object dominance index would equal 1, reflecting a total dominance of object identity information in the CNN response pattern for that layer. On the other hand, if a CNN layer responds to configuration changes only and is completely insensitive to object identity changes, then measure (1) would be zero and the index would equal -1, indicating a total configuration dominance in the CNN response pattern for that layer. In reality, the index would lie somewhere between -1 and 1, with a positive value indicating a bias in sensitivity towards object identity over configuration information and a negative value indicating the reverse.

We averaged the different Euclidean distance measures as well as the object dominance indices over the entire image collection for each sampled CNN layer and examined how these measures would evolve over the course of CNN processing and how this evolution would be affected by the different CNN architectures (Alexnet, Resnet18, Resnet50, and Densent161), spatial information contained in the background (present/absent), CNN classification accuracy for these scene images, and CNN training task (scene-recognition, object-recognition, or untrained). The results are reported in Figs 3 and 4.

We additionally conducted a number of statistical tests to evaluate the significance of the effects examined. Using the measures obtained from each image set as the dependent variable, we compared results from the "fixed" and "no background" image collections in repeated measure ANOVAs and tested for the effects of CNN layer, background type, and their interactions. We also used repeated measure ANOVAs to compare the scene-trained to the object-trained CNNs, as well as to compare the scene-trained to the untrained CNNs. To examine the effects in greater detail, we conducted pairwise t-tests for each layer, each time correcting for multiple comparisons for the number of layers sampled from a given CNN using the Benjamini-Hochberg method [28]. To quantify the overall trend (upward or downward) of the trajectory of the Euclidean distance and object dominance index measures over the course of processing, we correlated each measure against the rank order of the sampled layers.

To further quantify similarities between the object dominance index response profiles across changes in image sets and CNN training, for each CNN architecture, we reported the correlation of indices from all sampled layers between those obtained from the "fixed background" and "no background" collection, between those from the scene-trained and object-trained variants, and between those from the scene-trained and untrained variants. Because no formal correspondence exists between the sampled layers from different CNN architectures, we refrained from computing the correlations for indices obtained between different CNN architectures.

To understand how CNN scene classification accuracy may impact the object dominance index, based on the top 5 scene classification accuracy for each image set of the "variable background" image collection, we divided the collection into two halves, a high and a low accuracy half of image sets, each consisting of 10 image sets. We then obtained separate object

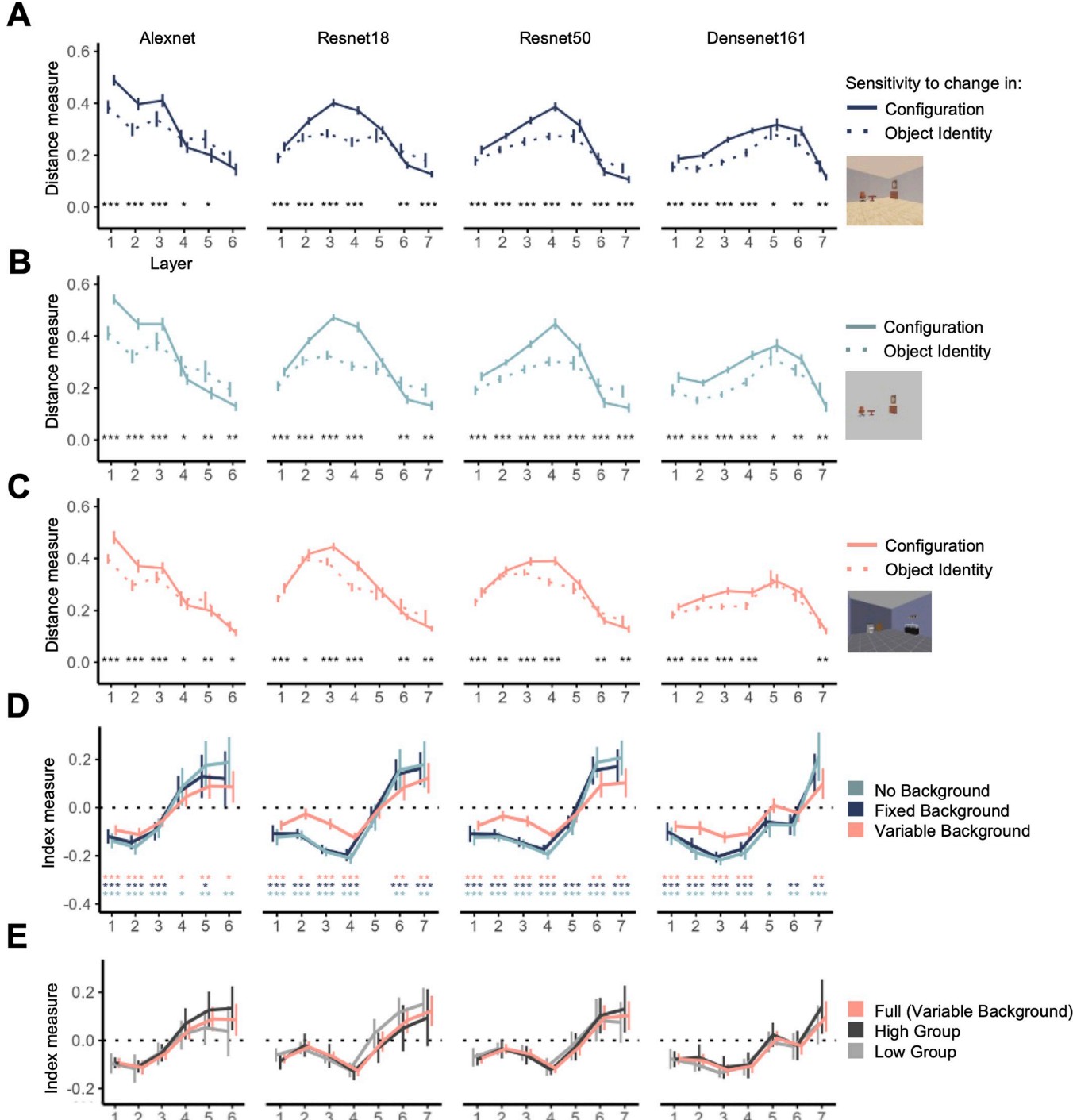

**Fig 3. Euclidean distance and index measures quantifying a scene-trained CNN's absolute and relative sensitivity to object identity and configuration changes for the different image collections.** The Euclidean distance for "Configuration" was calculated as the distance between two images sharing the same objects but different configurations. The Euclidean distance for "Object Identity" was calculated as the distance between two images sharing the same configuration but different objects. The object dominance index measured the relative sensitivity to object identity and configuration changes, with negative values indicating greater sensitivity to configuration than identity changes and positive values the reverse. **(A)** Euclidean distances for "Fixed Background" image collection. **(B)** Euclidean distances for "No Background" image collection. **(C)** Euclidean distances for "Variable Background" image collection. **(D)** The object dominance indices for all three image collections. **(E)** The object dominance indices for "Variable Background" image collection separated by scene classification accuracy. High/Low/Full Group, indices for the top-half/bottom-half/full image sets. Error bars indicate 95% confidence intervals of the means. In (A), (B), and (C), asterisks indicate significance from pairwise comparisons between the Euclidean distance measures using two-tailed t-tests (all corrected for multiple comparisons). All values are significantly non-zero *(ts > 10, ps < 001.)* In (D) the asterisks indicate the significance levels of the differences of each

measure at each sample CNN layer against zero using two-tailed *t* tests (all corrected for multiple comparisons). * $p < .05$, ** $p < .01$, *** $p < .001$. The plotting lines are slightly shifted horizontally with respect to each other to minimize overlap.

dominance indices for the two halves and reported them along with the index from the entire image collection in Fig 3.

Finally, to examine whether CNN sensitivity to configuration change is similar to general object position change, we compared the Euclidean distance and object dominance index measures from a configuration change to those from a uniform spatial translation in the "spatial shift" image collection. To measure sensitivity to translation, we calculated the Euclidean distance of CNN unit responses for images that differed in translation but shared the same object identity and configuration, similar to how sensitivity to a configuration change was calculated earlier. We also measured sensitivity to a configuration change as we did before with this image collection. We then derived both an object over configuration index as we did before, and an object over translation index. The latter is similar to how an object over configuration index is calculated, and is defined as $[(1') - (2')] / [(1') + (2')]$, in which $(1')$ is the Euclidean distance of unit responses for images sharing the same direction of translation (left/right) and configuration but different objects, and $(2')$ is the Euclidean distance of CNN unit responses for images that differed in translation but shared the same object identity and configuration. The Euclidean distance and index measures for translation were calculated separately for both configurations of each image set (see Fig 2) and then averaged together. Likewise, the Euclidean distances and index measures for configuration were calculated separately for both left and right shifted sets and then averaged together. These results are reported in Fig 5. To quantify the difference between these measures, we conducted repeated measure ANOVAs and pairwise t-tests by layer. As before, all t-tests were corrected for multiple comparisons for the number of layers included in a CNN using the Benjamini-Hochberg method [28].

To streamline the report of the statistical results, whenever possible, significance levels of the statistical results were reported directly in the figures and tables, rather than in the main text.

## Results

In this study, we measured the sensitivity of four CNNs to changes in the object identity and configuration information present in a scene image in order to understand how these two types of information contribute to scene representation at each stage of CNN processing. We additionally examined the effects of a structured 3D background, CNN classification accuracy, CNN architecture, and CNN training task on object identity and configuration representation during CNN scene processing. To understand whether CNNs differentiate between a relative object position change which occurs within a configuration change and any object position change, we additionally compared sensitivity to configuration change with that from a uniform spatial translation of the objects in a scene.

### The coding of object identity and configuration information in CNN scene representation and the effect of image background manipulation

To examine how object identity and configuration information may affect CNN scene representation, we first examined CNN responses to the "fixed background" image collection. This image collection consisted of 15 sets of scene images depicting the same indoor room background (see Fig 1 for examples). Each set contained four different images showing one of two object sets in one of two configurations. Each object set had four distinctive pieces of furniture.

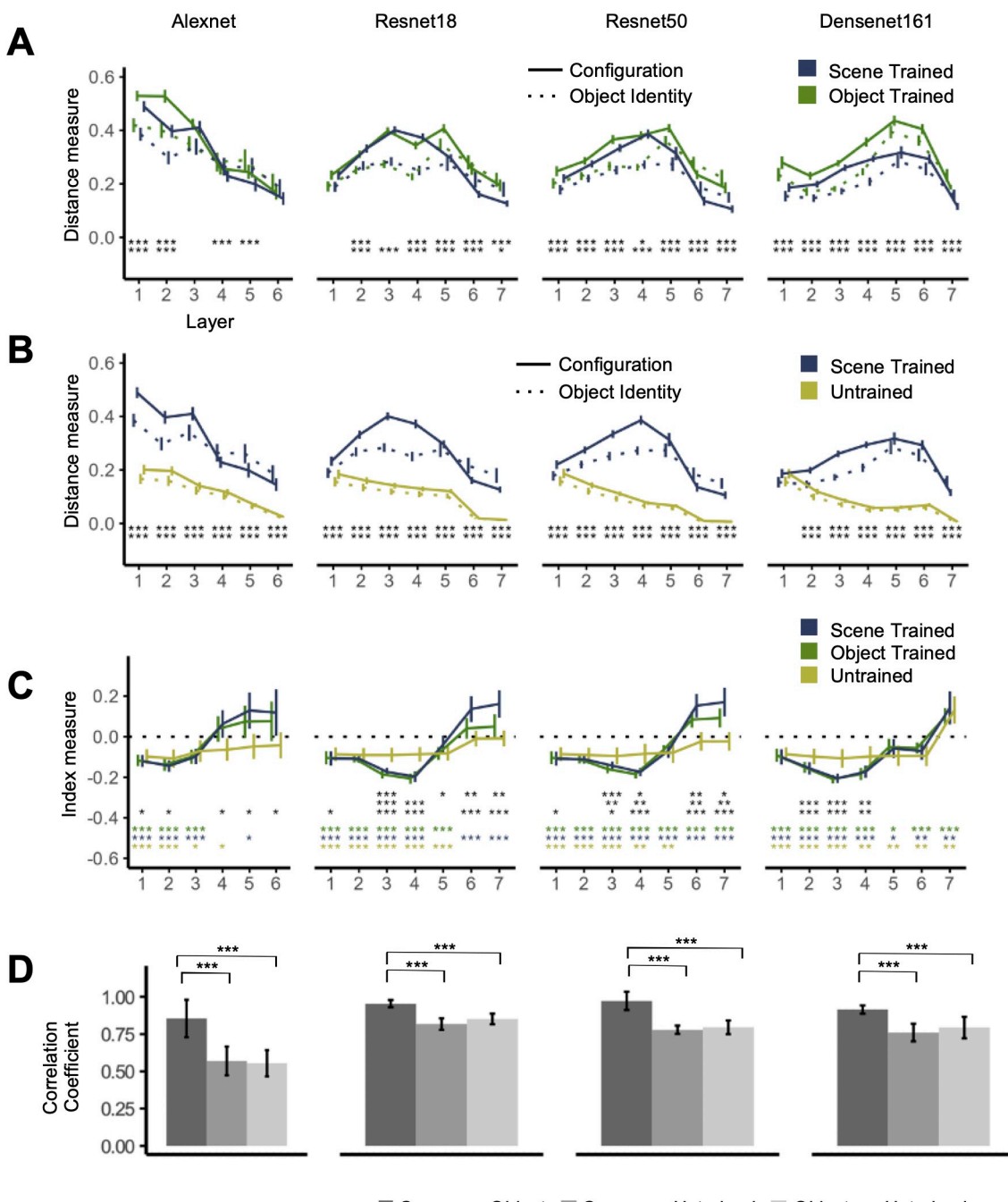

**Fig 4. Euclidean distance and index measures examining the effect of training, with these two types of measures quantifying a CNN's absolute and relative sensitivity to object identity and configuration changes, respectively.** The Euclidean distance for "Configuration" was calculated as the distance between two images sharing the same objects but different configurations. The Euclidean distance for "Object Identity" was calculated as the distance between two images sharing the same configuration but different objects. The object dominance index measured the relative sensitivity to object identity and configuration changes, with negative values indicating greater sensitivity to configuration than identity changes and positive values the reverse. (**A**) Euclidean distances for the scene-trained and object-trained CNNs. (**B**) Euclidean distances for the scene-trained and the untrained CNNs. (**C**) The object dominance indices for the different training regimes. (**D**) Pairwise correlations of the object dominance index curves across CNN layers for the three training regimes. The asterisks indicate the significance levels of the pairwise comparisons at each sample CNN layer using two-tailed *t* tests (all corrected for multiple comparisons). In (A) and (B), the top row is from comparing "Configuration" across the two types of training, and the bottom row is from comparing "Object Identity" across the two types of training. All values were significantly non-zero *(ts > 10, ps < .001)*. In (C), the asterisks from the top to bottom rows indicate, respectively, differences between scene-trained and object trained CNNs, between object-trained and untrained CNNs, between

scene-trained and untrained CNNs, between scene-trained CNNs and zero, between object-trained CNNs and zero, and between untrained CNNs and zero. Error bars indicate within-image sets 95% confidence intervals of the means. * $p < .05$, ** $p < .01$, *** $p < .001$. The plotting lines are slightly shifted horizontally with respect to each other to minimize overlap.

The two object sets contained the same four furniture categories but differed in the exact exemplars shown (e.g., two different couches appeared in the two object sets; see Methods for details). Each set thus contained two pairs of images with the same objects in different configurations and two pairs of images with different objects but the same configuration.

To investigate how strongly a CNN encodes object identity and configuration information in a scene image and how the coding strength varies over the course of processing, we obtained the averaged Euclidean distance between images sharing the same objects but different configurations and the averaged Euclidean distance between images sharing the same configuration but different objects. The results plotted against the sampled layers for each CNN are shown in Fig 3A. All statistical tests performed used the Euclidean distances from each image set as the dependent variable (i.e., the image sets provided the individual data points for the statistical tests with the error measures reflecting the error across the image sets).

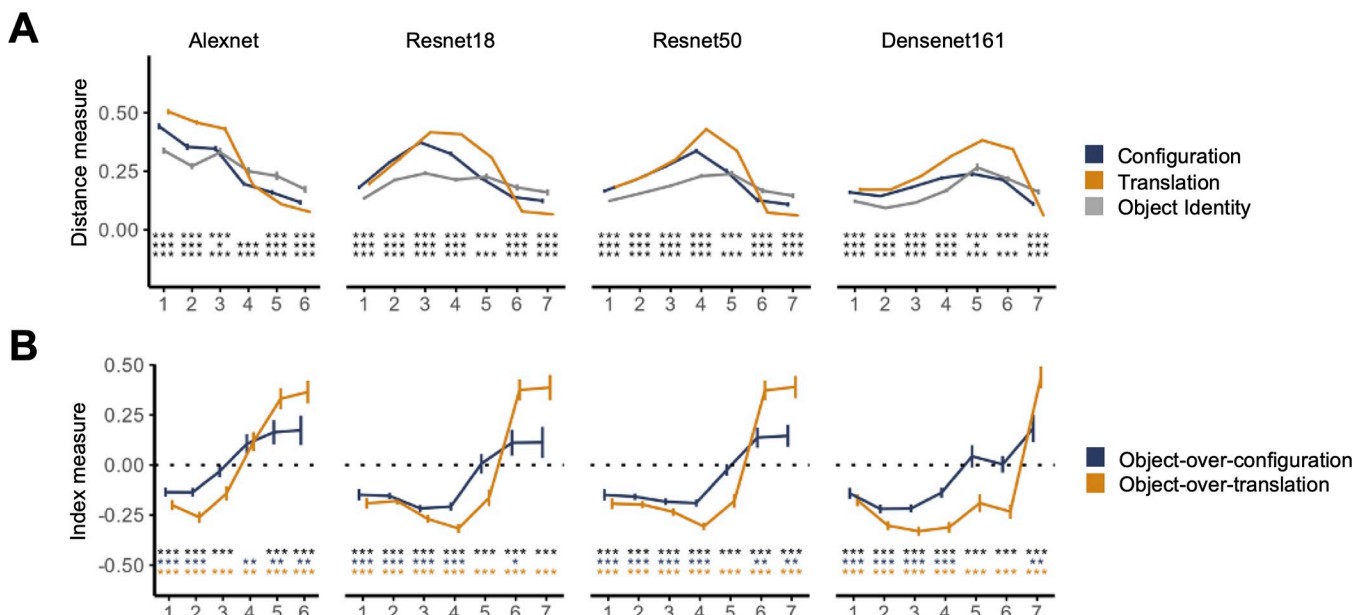

**Fig 5. Euclidean distance and index measures comparing the effect of configuration change and translation.** These two measures were obtained from scene-trained networks and quantify a CNN's absolute and relative sensitivity to object identity and configuration changes as well as left/right translation. (**A**) Euclidean distances for changes in configuration, translation and object identity. The Euclidean distance for "Configuration" was calculated as the distance between two images sharing the same objects and translation but different configurations. Euclidean distance for "translation" was calculated as the distance between two images sharing the same objects and configuration but different translation. The Euclidean distance for "Object Identity" was calculated as the distance between two images sharing the same configuration and translation but different objects. (**B**) The object dominance indices for configuration change and translation. The object-over-configuration indices measured the relative sensitivity to object identity and configuration changes, with negative values indicating greater sensitivity to configuration than identity changes and positive values the reverse. The object-over-translation indices measured the relative sensitivity to object identity and translation changes, with negative values indicating greater sensitivity to translation than identity changes and positive values the reverse. The asterisks indicate the significance levels of the pairwise comparisons made at each sampled layer using two-tailed $t$ tests (all corrected for multiple comparisons). In (A), the asterisks from the top to bottom rows indicate, respectively, differences between "Configuration" and "Translation", between "Configuration" and "Object Identity", and between "Translation" and "Object Identity." In (B), the asterisks from the top to bottom rows indicate, respectively, differences between the two index measures, differences between the object-over-configuration indices and zero, and differences between the object-over-translation indices and zero. Error bars indicate within-image sets 95% confidence intervals of the means. * $p < .05$, ** $p < .01$, *** $p < .001$. The plotting lines are slightly shifted horizontally to minimize overlap.

**Table 3. Correlation coefficients of Euclidean distance and index measures with the rank order of the sampled CNN layers from each scene-trained CNNs for the different image collections.**

| CNN | Image set collection | | |
|---|---|---|---|
| | **Fixed Background** | **No Background** | **Variable Background** |
| Alexnet | -0.96 (-40.91/***) | -0.96 (-57.93/***) | -0.97 (-39.26/***) |
| | -0.86 (-13.54/***) | -0.86 (-12.91/***) | -0.91 (-21.08/***) |
| | **0.79 (8.66/***)** | **0.86 (12.27/***)** | **0.81 (10.69/***)** |
| Resnet18 | -0.56 (-19.80/***) | -0.59 (-22.72/***) | -0.71 (-29.31/***) |
| | -0.22 (-2.18/***) | -0.35 (-4.93/***) | -0.58 (-10.82/***) |
| | **0.72 (14.93/***)** | **0.67 (8.22/***)** | **0.63 (8.72/***)** |
| Resnet50 | -0.47 (-15.16/***) | -0.45 (-13.67/***) | -0.63 (-19.38/***) |
| | -0.20 (2.76/***) | -0.09 (-1.00/ns) | -0.54 (-10.48/***) |
| | **0.74 (15.22/***)** | **0.76 (20.79/***)** | **0.61 (7.69/***)** |
| Densenet161 | 0.03 (0.77/ns) | -0.07 (-1.22/ns) | -0.25 (-4.00/***) |
| | 0.44 (7.81/***) | 0.47 (7.42/***) | 0.07 (0.79/ns) |
| | **0.60 (7.74/***)** | **0.62 (7.18/***)** | **0.6 (9.68/***)** |

First two rows of each CNN are for the Euclidean distance measures, "Same Objects/Different Configuration" and "Different Objects/Same Configuration", respectively. The bolded third row of each CNN is for the index measures. Coefficients ($r$) were calculated separately for each image set and then averaged over all the image sets in a collection. The coefficients were tested at the group level against zero using two-tailed $t$ tests. Results include the r value, the t-value, and p-value in [r/(t/p)] format. *** $p < .001$, ns $p > .05$.

Across all CNNs we examined, all sampled layers showed significant non-zero Euclidean distance measures for both object identity and configuration changes ($ts > 10\ ps < .001$; all corrected). This indicates that all CNNs were sensitive to changes in both object identity and configuration information throughout each stage of processing.

To assess how sensitivity to object identity or configuration change may evolve throughout CNN scene processing, we correlated each of these two Euclidean distance measure against the rank order of the sampled layer for each image set and averaged over all image sets to produce an averaged correlation coefficient for the entire image collection. A positive/negative correlation would indicate that a CNN becomes more/less sensitive to the object identity or configuration changes over the course of scene processing. For both types of Euclidean measures, we found strongly negative correlations in Alexnet, moderately negative correlations in Resnet18 and Resnet50, and a moderately positive correlation in Densenet161 (see Table 3 for the full statistical results). Overall, it appears that different CNN architectures differ in their absolute sensitivity to object identity and configuration changes over the course of scene processing.

For each CNN, we found that sensitivity to object changes is distinct from that of the configuration changes across the CNN layers. Directly comparing the two Euclidean distance measures in a repeated measure ANOVA test, we found a significant or marginally significant main effect of Euclidean distance type, and importantly, a highly significant interaction of Euclidean distance type with layer in all CNNs (see Table 4 for the full stats). Pairwise t-tests further indicated that all CNNs responded more strongly to configuration changes than object identity changes in the early layers, but that the reverse was true in the final layers (see the asterisks marking the significance values on Fig 3A; all corrected).

To better characterize the relative sensitivity of a CNN to object identity vs configuration changes, and how this relative sensitivity may evolve across different stages of the processing, we defined an *object dominance index* (see Methods; see also [13]). This index varies between -1 to 1, with a positive value indicating a greater sensitivity for object identity over configuration changes, and a negative value indicating the reverse. Like the Euclidean distance

**Table 4. Statistical results of repeated measure ANOVA tests comparing the two types of Euclidean distance measures (Different Object/Same Configuration, Same Object/Different Configuration).**

| Network | Image Collection | | | Training Regime | | |
|---|---|---|---|---|---|---|
| | Fixed Background | No Background | Variable Background | Scene | Object | Untrained |
| Alexnet | 466.64/*** | 450.84/*** | 546.99/*** | 466.64/*** | 315.91/*** | 492.74/*** |
| | 3.82/*ns* | 2.59/*ns* | 5.05/* | 3.82/ns | 7.28/* | 13.12/** |
| | 55.36/*** | 98.25/*** | 77.56/*** | 55.36/*** | 57.35/*** | 29.80/*** |
| Resnet18 | 243.31/*** | 331.47/*** | 278.42/*** | 243.31/*** | 102.95/*** | 496.40/*** |
| | 30.72/*** | 40.64/*** | 5.42/* | 30.72/*** | 48.13/*** | 31.66/*** |
| | 118.97/*** | 119.17/*** | 59.66/*** | 118.97/*** | 119.99/*** | 34.55/*** |
| Resnet50 | 281.24/*** | 239.55/*** | 258.77/*** | 281.24/*** | 84.49/*** | 424.60/*** |
| | 33.45/*** | 42.03/*** | 9.46/** | 33.45/*** | 52.39/*** | 24.33/*** |
| | 159.11/*** | 171.72/*** | 65.17/*** | 159.11/*** | 158.62/*** | 24.63/*** |
| Densenet161 | 241.72/*** | 149.54/*** | 90.91/*** | 241.72/*** | 217.72/*** | 383.25/*** |
| | 35.95/*** | 38.44/*** | 14.25/** | 35.95/*** | 55.68/*** | 26.18/*** |
| | 59.19/*** | 46.83/*** | 47.30/*** | 59.19/*** | 51.28/*** | 32.71/*** |

The three rows of each CNN report the main effect of layer, main effect of Euclidean distance type, and the interaction between layer and Euclidean distance type respectively. Results include the F-value and p-value in (F/p) format.

*** $p < .001$

** $p < .01$

* $p < .05$, ns $p > .05$.

measures, this index was calculated separately for each of the image sets and for each sampled CNN layer, and the results were averaged across all image sets to generate an average object dominance index for that layer.

Fig 3D shows the average object dominance indices plotted against sampled layers for each CNN. Through repeated measure ANOVA testing, we found a significant effect of layer for each CNN ($Fs > 30$, $ps < .001$). Moreover, all CNNs showed significantly negative object dominance indices in earlier layers which become positive in the final layer (see the asterisks marking the significance level of the t-tests on Fig 3D; all corrected; note that the effect was marginally significant for the final layer of Alexnet at $p = 0.051$ and significant for all other CNNs). Additionally, in all CNNs, the first layer demonstrated a lower index measure than the last layer, and there was a significantly positive correlation between the index measure and the rank order of the sampled layer (see Tables 3 and 5 respectively for the detailed stats). Thus, despite differences in both architecture and absolute sensitivity to object identity and configuration changes over the course of processing, all CNNs exhibited an upward trending object

**Table 5. Paired, one-tailed t-test results comparing differences in index measures between the first and final layer.**

| Network | Image Collection | | | Training Regime | | |
|---|---|---|---|---|---|---|
| | Fixed Background | No Background | Variable Background | Scene | Object | Untrained |
| Alexnet | -4.54/*** | -7.02/*** | -6.46/*** | -4.54/*** | -4.47/*** | -2.64/** |
| Resnet18 | -7.99/*** | -6.76/*** | -7.70/*** | -7.99/*** | -5.84/*** | -5.00/*** |
| Resnet50 | -8.21/*** | -9.51/*** | -7.25/*** | -8.21/*** | -7.53/*** | -3.62/** |
| Densenet161 | -6.94/*** | -6.93/*** | -6.77/*** | -6.94/*** | -8.81/*** | -7.72/*** |

Results include the t-value and p-value in (t/p) format.

*** $p < .001$

** $p < .01$.

dominance index, with early layers showing more sensitivity to configuration than object identity changes and later layers showing the reverse.

To understand if our results depended on the presence of spatial information present in the image background, we also created a "no background" image collection by removing the 3D room scene of each image in the "fixed background" image collection (see Fig 1C for examples). We found some differences accompany the background change, including, for both Euclidean distance measures a significant interaction of background with layer in all CNNs ($Fs > 3.2$, $ps < .007$), and for the index measure, a significant interaction between background and layer in all CNNs except for Resnet18 ($F = 1.7$, $p = .13$ for Resnet18; $Fs > 4.1$, $ps < .001$ for all others).

However, the differences introduced by removing the 3D room scene background are relatively small and the overall shape of the response pattern is preserved for the index measures (Fig 3B). Specifically, each CNN still exhibited significantly negative object dominance indices in early layers and significantly positive object dominance indices in the final layers (see the asterisks marking the significance level of the t-tests on Fig 3D; all corrected). Moreover, we continue to see in all the CNNs a lower index in the first than in the final layer as well as significantly positive correlations between the index measure and layer (see Tables 3 and 5 respectively for the detailed stats).

This similarity is further exemplified in a high correlation between the object dominance indices for the "fixed background" collection and the "no background" collection across layers ($rs > .83$, $ts > 7.6$, $ps < .001$). These results indicate that the presence of a spatially structured background does not significantly impact the overall profile of a CNN's relative sensitivity to object identity vs configuration changes over the course of information processing.

## Effects of CNN classification accuracy

Although all CNNs examined here exhibited high sensitivity to object identity and configuration changes in the computer-generated images used here, it may be argued that CNNs may not classify these images accurately and that how CNNs represent scenes depends on their success in scene classification. To address this question, we first looked at the classification accuracy of the images in the fixed background and no background collections. Despite a markedly worse classification accuracy for images with no backgrounds than those with fixed backgrounds (see Table 2 for the classification accuracy results), we still saw a high degree of similarity in the response profiles as reported in the last section of the results. Thus, how a CNN represents object identity and configuration in a scene is not strongly affected by how well the CNN can classify the scene image.

To more directly test the effects of classification accuracy, we also examined responses from the "variable background" image collection which contained 20 image sets. Each image set in this collection had a unique and object-matching scene background (e.g., a tiled wall for a bathroom scene) to more closely mimic the natural scenes used to train the CNNs (see Fig 1D for examples). Including all the image sets in this collection, we were able to replicate our findings with the fixed and no background image collections. As before, for the distance measures, we found that for all CNNs, each layer exhibits significantly positive Euclidean distance measures ($ts > 10$, $ps < .001$; all corrected), with a main effect of the Euclidean distance type and an interaction of distance type with layer (see Table 4 for the full stats); for the index measures, we again found a significantly negative index in the earlier layers which became significantly positive in the final layers (see Fig 3D for the asterisks marking the t test results, all corrected), with the index being lower at the earlier than later layers, and a significant positive correlation between the index measure and the rank order of the sampled layer (see Tables 3 and 5 respectively for the full stats).

To more closely examine the effect of classification accuracy on representation, we divided this image collection into a high and a low classification accuracy group based on the classification accuracy. We repeated our analysis within each group and compared the resulting object dominance indices between the two groups. We found overall high correlations ($rs > .97$) between the two accuracy groups and each with the full group (Fig 3E). This again shows that the overall profile of how a CNN represents object identity and configuration information in a scene image remains relatively stable regardless of whether or not the CNN can classify that image accurately.

Our results thus show that the ability of a CNN to correctly classify a scene image does not significantly impact how it weights object identity and configuration information in a scene. We continued to observe a persistent pattern whereby CNNs are more sensitive to object configuration than identity changes during early stages of processing, gradually lose this bias and become more sensitive to identity than object configuration changes at the final stages of their processing.

## Effects of training on CNN object identity and configuration representation

Prior research has shown that CNNs trained on a scene recognition task (scene-trained CNNs) were more accurate at classifying scene images than those trained on an object recognition task (object-trained CNNs) [7]. Thus, training a network to classify scenes may increase its selectivity for scene relevant information, such as the configuration of objects in the scene. This would predict that scene-trained CNNs would exhibit a great sensitivity to changes in configuration information, and therefore a more negative object dominance index than the corresponding object-trained CNNs. Alternatively, given that CNN's scene classification accuracy does not affect the index profile as reported above, changing the training task may minimally impact object and configuration representation in CNNs. To investigate these two possibilities, we compared object and configuration representations of the "fixed background" image collection in object-trained and scene-trained CNNs sharing the same architecture.

Similar to the scene-trained CNNs, the object-trained CNNs show significant sensitivity to both object identity and configuration changes ($ts > 10$, $ps < .001$; all corrected). Although a repeated measure ANOVA reveals significant effect of training and an interaction between training with layer on both types of Euclidean distance measures in all CNNs ($Fs > 11$, $ps < .005$), detailed comparisons do not reveal any clear patterns. Specifically, object-trained CNNs are not consistently more sensitive to object identity changes, nor are scene-trained CNNs consistently more sensitive to configuration changes (see the asterisks marking the significance values on Fig 4A; all corrected). In fact, in Resnet50 and Densenet161, the object-trained CNN is more sensitive to configuration changes than the respective scene-trained CNN. Correlation of each Euclidean distance measure to the rank order of the sampled layers also revealed subtle differences but no overall patterns (see Table 6).

For the index measures, using repeated measure ANOVAs, we found a main effect of training in Resnet18 and Resnet50 ($Fs > 11$, $ps < .006$) but not in Densenet161 ($F = .17$, $p = .69$) or Alexnet ($F = .88$, $p = .36$). We additionally found an interaction of training with layer in Resnet18, Resnet50, and Densenet161 ($Fs > 2.3$, $ps < .036$), but not in Alexnet ($F = 1.6$, $p = .19$). A series of pairwise t-tests between the indices for the scene- and object-trained CNNs revealed that for all CNN architectures, the object dominance indices of the scene-trained CNN were either greater or not significantly different from those of the object-trained CNN (see the asterisks marking the significance values on Fig 4C; all corrected). We therefore failed to find evidence showing that scene-trained CNNs would exhibit a great sensitivity to

**Table 6. Correlation coefficients of Euclidean distance and index measures with the rank order of the sample CNN layers from scene-trained, object-trained and untrained CNNs for the fixed background image collections.**

| CNN | Scene | Object | Untrained |
|---|---|---|---|
| Alexnet | -0.96 (-40.91/***) | -0.96 (-62.33/***) | -0.98 (-33.85/***) |
| | -0.86 (-13.54/***) | -0.95 (-23.43/***) | -0.98 (-22.30/***) |
| | **0.79 (8.66/***)** | **0.84 (12.03/***)** | **0.63 (5.09/***)** |
| Resnet18 | -0.56 (-19.80/***) | -0.17 (-2.84/*) | -0.93 (-43.70/***) |
| | -0.22 (-2.18/*) | 0.24 (2.73/*) | -0.93 (-46.19/***) |
| | **0.72 (14.93/***)** | **0.60 (10.46/***)** | **0.69 (7.80/***)** |
| Resnet50 | -0.47 (-15.16/***) | -0.20 (-3.26/**) | -0.98 (-56.00/***) |
| | -0.20 (-2.76/*) | 0.29 (2.66/*) | -0.98 (-61.62/***) |
| | **0.74 (15.22/***)** | **0.70 (14.01/***)** | **0.64 (5.29/***)** |
| Densenet161 | 0.03 (0.77/*ns*) | 0.19 (4.60/***) | -0.90 (-98.21/***) |
| | 0.44 (7.81/***) | 0.49 (8.29/***) | -0.89 (-97.38/***) |
| | **0.60 (7.74/***)** | **0.62 (9.96/***)** | **0.54 (8.22/***)** |

Each of the untrained CNNs was the average of 100 random weight initialized CNNs. The first two rows of each CNN are Euclidean distance measures of "Same Object/Different Configuration" and "Different Object/Same Configuration" respectively. The bolded third row of each CNN are the index measures. Coefficients (*r*) were calculated separately for each image set and then averaged over all the image sets in a collection. The coefficients were tested at the group level against zero using two-tailed *t* tests and the resulting *t* and *p* values are reported in parentheses. Results include the r value, the t-value, and p-value in [r/(t/p)] format.

*** p < .001

** p < .01

* *p < .05.*

configuration changes or a more negative object dominance index than the corresponding object-trained CNNs.

In fact, we still found the profile of the index responses to be remarkably similar between the two. Specifically, in all CNNs, we continued to see both types of distance measures to be greater than zero in all the sample layers (*ts* > 10, *ps* < .001; all corrected), and the presence of both a main effect of the Euclidean distance type and an interaction of distance type with layer (see Table 4 for the full stats). Likewise for index measures, as in scene-trained CNNs, in the object-trained CNNs, we noted negative indices in the early layers and positive indices in the final layer (though in Alexnet and Resnet18, the indices were not different from zero; see the asterisks marking the significance values on Fig 4C; all corrected). In all CNNs, we observed a lower index at the beginning of the convolution than at the end, and we found positive correlations between the index and the rank order of the sampled layers in all the CNNs (see Tables 5 and 6 respectively for the detailed stats).

To further understand the effects of the training task on the sensitivity of a CNN to changes in object and configuration information, we compared the responses obtained from the scene-trained CNNs to those obtained from untrained CNNs. To obtain the untrained CNN data for a given CNN architecture, we randomly initialized the weights of the CNN and extracted its responses to the "fixed background" collection of scene images. We repeated this process 100 times and then averaged the Euclidean distance and index measures, to obtain the respective measures for the untrained CNN.

The untrained CNNs showed sensitivity to both object and configuration changes, exhibiting significantly positive Euclidean distance measures across all the sampled layers (*ts* > 26, *ps* < .001; all corrected). Notably, after the first layer, all trained CNNs demonstrated significantly

greater Euclidean distance measures and hence greater absolute sensitivity to both object identity and configuration changes than their untrained counterparts (see the asterisks marking the significance levels in Fig 4B; all corrected). For all untrained CNNs, both Euclidean distance measures demonstrated strongly significant negative correlations to the rank order of the sampled layer (see Table 6). This indicates that the absolute sensitivity to both object and configuration changes decreases over the course of processing in these CNNs. Untrained CNNs are therefore both less sensitive to changes in object identity and configuration information, but also less able to propagate information about these changes within their internal representation from one layer to the next.

In the untrained CNNs, as in the trained CNNs, we nevertheless found a main effect of Euclidean distance type and interactions between distance type and sampled layer (see Table 4 for detailed stats), lower index measure in the first than in the final layer, and positive correlations of index measures with layer numbers in all CNNs (see Tables 5 and 6 respectively). Meanwhile, differences in index profiles were also observed for the trained and untrained CNNs (Fig 4B). Direct comparison between the trained and untrained CNNs using a repeated measure ANOVA revealed no main effect of the presence of training ($F = 3.9$, $p = .066$ in Alexnet; $Fs < 2.4$, $ps > .14$ in all other CNNs) but a highly significant interaction between layer and training in all four CNNs ($Fs > 11$, $ps < .001$). Pairwise comparisons by layer reveal that trained and untrained CNNs demonstrate similar indices at the beginning of the processing, but that training increased sensitivity to configuration change in the middle layers of Resnet18, Resnet50, and Densenet161, and increased sensitivity to object identity changes in the final layers of Alexnet, Resnet18, and Resnet50 (see the asterisks marking the significance levels in Fig 4C; all corrected). Moreover, across all untrained CNNs, except for the final layer of Densenet161 which exhibited a positive index, the indices in all other layers were either negative or were not statistically different from zero (see the asterisks in Fig 4C).

Comparing the index profiles across layers of the object-trained, scene-trained, and untrained CNNs, we found that scene-trained and object-trained CNNs showed a higher correlation with each other than they did with the untrained CNNs (see Fig 4D and the asterisks marking the significance levels), indicating an effect of training on the index profile.

Overall, these results indicated that training affected how object identity and configuration information was represented, with different CNN architectures expressing this effect in different ways. Considering the relative sensitivity of object identity vs configuration information, while the exact values might differ, the overall response profile was similar between scene and object trained CNNs, with earlier layers preferring configuration and later layers preferring object information. The presence of training itself had a significant effect on the sensitivity of a CNN to both object identity and configuration changes, with the trained CNNs demonstrating a significantly greater sensitivity to both types of changes compared to untrained CNNs. As the index measures from both trained CNNs were more correlated to each other than to untrained CNNs, it appeared that although untrained CNNs shared certain similarities in the index coding profile, there was an additional effect of training on the relative coding strength of object and configuration information in all the CNNs examined here.

## Comparing configuration change with uniform spatial translations in scenes

Our configuration manipulation required objects to change relative positions across two images. To understand whether object position changes from a configuration change were represented differently from any type of position changes in scene-trained CNNs, such as uniform left and right spatial translations, we directly compared these two types of position changes in

the "spatial shift" image collection. This image collection contained the size-reduced images from the "no background" image collection and shifted all objects in an image horizontally either to the left or right (see Fig 2). This allowed us to examine the effect of object translation without an object identity and configuration change, as measured by the Euclidean distance between two images sharing the same objects and configurations but differing in the direction of translation. As in the previous analysis, we obtained Euclidean distance measures for object identity and configuration changes by holding absolute position (translation) constant. Besides obtaining an object-over-configuration index as we did before, using a similar procedure, we also constructed an object-over-translation index (see Methods). The resulting Euclidean distance and index measures are shown in Fig 5A and 5B, respectively.

For all CNNs and all sampled layers, all three Euclidean distance measures were greater than zero, showing that all the CNNs exhibited sensitivity to changes in identity, configuration, and translation to these images. Repeated measures ANOVA revealed main effects of both layer and type of Euclidean distance measure, as well as an interaction between the two in all four CNNs ($Fs > 5.80$, $ps < .008$). Pairwise t-tests revealed that the magnitudes of the three Euclidean distance measures varied over the course of processing. Translations tended to be higher than the other two measures in the earlier and mid layers, but becoming lower at the end of processing (see the asterisks marking the significance levels in Fig 5A; all corrected). For the index measures, repeated measures ANOVA revealed an effect of layer as well as an interaction between layer and the type of position change in all four CNNs ($Fs > 77.98$, $ps < .001$). There was also a main effect of the type of position change in Densenet161 ($Fs > 70.80$, $ps < .001$) but not in the other CNNs ($Fs < 3.56$, $ps > .08$). Pairwise t-tests revealed that the object-over-translation index was more negative than the object-over-configuration index in the first few layers of all four CNNs (Fig 5B), indicating a higher sensitivity to translations than configuration changes in early stages of CNN processing. By the end of CNN processing, the pattern reversed in all four CNNs such that the object-over-translation index was greater than the object-over-configuration index (see the asterisks marking the significance values in Fig 5B; all corrected). This indicated a stronger sensitivity to an object configuration change than to a translation by the end of CNN processing. The correlation of each measure with the rank order of the CNN layers sampled is reported in Table 7.

These results show that a CNNs' sensitivity to a configuration change was not the same as their sensitivity to any type of position change, such as that induced by a uniform spatial translation of the objects without a configuration change. Even though all CNNs showed a greater

**Table 7. Correlation coefficients of Euclidean distance and index measures comparing configuration and uniform translation manipulations.**

| Network | Distance Measure | | | Index | |
|---|---|---|---|---|---|
| | Same Obj Diff Config | Same Obj Diff Spatial | Diff Obj Same Config | Object-Configuration | Object-Translation |
| Alexnet | -0.96/*** | -0.96/*** | -0.81/*** | 0.94/*** | 0.88/*** |
| Resnet18 | -0.49/*** | -0.51/*** | -0.01/ns | 0.74/*** | 0.66/*** |
| Resnet50 | -0.34/*** | -0.36/*** | 0.18/* | 0.75/*** | 0.78/*** |
| Densenet161 | 0.05/ns | 0.11/*** | 0.62/*** | 0.59/*** | 0.80/*** |

The correlation coefficients were calculated by correlating the measures with the rank order of the sampled CNN layers from the scene-trained CNNs for the no background image collections. Coefficients (r) were calculated separately for each image set and then averaged over all the image sets in a collection. The coefficients were tested at the group level against zero using two-tailed *t* tests and the resulting significance were reported. Results include the r-value and p-value in (r/p) format.

\*\*\* p < .001

\*\* p < .01

\* p < .05, ns p > .05.

sensitivity to translation at early stages of processing, likely due to a greater amount of position change in translation than in the configuration change, by the end of processing, the pattern reversed with all CNNs showing a greater sensitivity to the configuration change. A change in the relative positions among the objects was thus distinct from an otherwise similar, if not greater, amount of positional change induced by moving all the objects together while preserving the relative positions among the objects.

## Discussion

Despite decades of cognitive and neuroscience research on visual scene representation, presently our knowledge is still incomplete regarding how object identity and configuration information is weighted during scene processing and how this weighting evolves over the course of scene processing. Recent developments in CNN modeling have enabled these artificial visual processing systems to achieve high performance in scene classification tasks [7, 8]. Importantly, responses from CNNs during scene processing have been shown to correlate with neural responses from human scene processing regions ([9]; see also [10–12]). At the same time, because CNNs are extremely complex models with millions, if not more, free parameters, the general operating principles at the algorithmic level (see [17]) supporting CNNs' success in object and scene categorization remain largely unknown (e.g., [18]). The mere correspondence in representation between the brain and CNNs in scene processing thus does not provide us with a deeper understanding of how scene processing is done in the human brain. Nevertheless, because CNNs can be tested much easier than running expensive neuroimaging experiments on human participants, given the correspondence between the brain and CNNs, studying the details of scene processing in CNNs not only can enrich our understanding of the computational algorithms governing scene processing in CNNs, but could also provide guidance as to how to study and understand scene processing in the human brain.

In this study, we measured the sensitivity of CNNs to changes in object identity and configuration information in a scene image over the course of processing. We consider both the absolute and the relative strength of this sensitivity to both types of changes. Overall, we found variations in each CNN's absolute sensitivities to object identity and configuration changes over the course of scene processing, with sensitivity declining in Alexnet, somewhat fluctuating in Resnet18 and Resnet50, and generally increasing in Densenet161. Despite these differences, when the relative sensitivity to object identity over configuration information is considered, a common pattern emerges. For all CNN architectures, we observed a shared coding profile characterized by configuration representation dominating object identity representation in early layers and the reverse being true in the final layers. Because CNNs are trained for scene classification only and because a reasonable configuration change of the objects in a scene does not change its classification, CNNs are not required to code such object configuration. Our finding that CNNs demonstrate sensitivity to configuration changes suggests that objects and configurations are likely fundamental elements of scene representation, such that a system capable of classifying scenes would automatically encode both elements.

Interestingly, within each CNN architecture, we observed the profile of the object dominance index to persist even when we removed the spatial structure from the scene background, and despite a marked drop in the classification accuracy of the CNN. Because the CNN training set contains natural scene images with backgrounds present, CNNs expectedly learn that backgrounds are an essential part of a scene. Thus, when such backgrounds are removed, CNNs may no longer assign the same label to the scene. Label assignment occurs at the very end of CNN scene processing. As such, up to this last stage, CNNs should be blind as to whether or not a scene would match one of its pre-learned labels. In this regard, CNN scene

processing should not differentiate between scenes it can and cannot label. What is most critical here is whether a CNN would show sensitivity to object and configuration changes throughout the course of CNN processing. Our results show that this was indeed the case whether or not scene background was present.

Both scene-trained and object-trained CNNs exhibited similar responses in their relative coding strength of object and configuration information over the course of processing. This could be due to either the shared network architecture, a shared processing demand, or both. Due to shared network architecture, scene-trained and object-trained CNNs could utilize a common processing algorithm to perform both types of classification, with training providing the learning of the relevant input information for each task. Alternatively, because both a scene and an object may contain configurations of more basic elements, i.e., a room scene with a particular arrangement of furniture and a complex object with a specific configuration of parts, the similar demand of scene and object classification may lead to the development of a convergent processing algorithm for both.

Our results indicate that the response profile for the object dominance index from untrained CNNs (CNNs with randomly initialized weights) exhibited some similarity to those found for the trained CNNs. This suggests that some of the characteristic dominance of configuration information in early stages of processing and the reversal to object identity dominance in later layers may arise as an effect of intrinsic network architecture. In the untrained Alexnet, Resnet18, and Resnet50, we observed that the object dominance index, initially negative, tends towards zero over the course of processing. Considering the Euclidean distances from these trials, we observed a gradual decline of both measures that is strongly correlated with the rank order of sampled layers (see Table 6). This is consistent with the idea that due to downsampling and pooling, information about the objects and configuration in the original scene image is gradually lost.

However, we do not believe that the effects of pooling and downsampling could account for the entirety of our results. For the Euclidean distance measures, the scene-trained Resnet18, Resnet50, and Densenet161 demonstrated more complex trajectories, and between consecutive sampled layers, there was even an increase in the distance measures (see Fig 4B). In these trained CNNs, information about object and configuration changes is thus not only retained but enriched, in contrast to the monotonic decline of the same distance measures observed in the untrained CNNs. (see Table 6). For the object dominance index measures, the final layer of scene-trained Alexnet, Renset18, and Resnet50 all showed significantly positive indices that were greater than the respective untrained CNN (see Fig 4C). These results indicate that training exerts a significant additional effect on both the Euclidean distance and the object dominance measures.

Even though training increased position tolerance over the course of CNN processing (as evident by the finding that object changes are represented more strongly than a large uniform spatial translation over the course of CNN processing), CNNs' sensitivity to a configuration change was not the same as their sensitivity to any type of position change, such as that induced by a uniform spatial translation of the objects without a configuration change. By computing both the object-over-configuration and object-over-translation indices, we noted that in relation to the same object change, each CNN demonstrated a greater sensitivity towards a uniform translation than configuration change at earlier stages of processing. If configuration could be trivially explained as a position change, then these results would predict that CNNs would continuously exhibit less sensitivity to a configuration change (due to its smaller absolute position change magnitude) than to a uniform translation over the entire course of CNN processing. Yet, towards the end of processing, we observed the exact opposite, with CNNs showing more sensitivity to a configuration change than a translation. This

suggests that configuration is a distinct type of position change from a uniform spatial translation. That being said, it would be important for future studies to help us understand whether configuration is explicitly represented or whether it is implicitly coded among the objects in a scene. One manipulation could be to change the viewpoint of the scene. This would change some relative positions among the objects without changing the true configuration among the objects. Another manipulation could be to use a schematic representation of the configuration (e.g., by replacing all furniture in a room with cubes) and test its similarity with the actual room scene sharing the same configuration.

In the human brain, both fMRI and single neuron responses in human scene region PPA exhibited stronger responses to images with spatial layout than those without [1, 29]. In CNNs, we found here that removing the spatial structure of the background had only minor effects on the sensitivity of a CNN to either object identity or configuration changes as well as the relative coding strength of these two types of information, though it did markedly decrease overall classification accuracy (for the reasons stated earlier). It is possible that the arrangements of the furniture in our images sufficiently depicted a 3D spatial layout and resembled the real-world scenes used for CNN training. It would be interesting in future studies to scramble the 3D arrangement of the furniture following the procedure used in [1] and test if a similar response profile to object identity and configuration changes can still be obtained in CNNs.

Besides scene processing regions in the human brain, higher visual object processing regions in the human lateral occipital complex (LOC) [30–32] have also been shown to be involved in encoding scene information [33, 34]. Whereas the PPA and RSC have been shown to respond to background spatial structure, scene processing responses in the LOC were not found to be modulated by the presence of a spatially structured background [35]. While scene processing differences clearly exist in the human brain between object and scene-processing regions, both scene-trained and object-trained CNNs exhibited similar responses in the present study. This is consistent with the finding of [12] who also showed that the middle layer of an ImageNet trained VGG-S network showed the highest correlation with both LOC and PPA. It would be interesting in future studies to present human participants the images used here and measure their fMRI responses in both object- and scene-selective regions of their brains and examine whether a similar response profile is present in both regions of the human brain. Doing so would not only enrich our understanding of object and scene processing in the human brain, but also identify potential processing differences between the human brain and CNNs.

To conclude, our results provide one of the first documentations of how object identity and configuration information is weighted during scene processing in CNNs and how this weighting evolves over the course of scene processing, thereby providing a deeper understanding of the nature of visual representation in CNNs. Given the documented correspondence between the brain and CNNs, our results may provide clues regarding the potential computational algorithms employed by the human brain in visual scene processing. The experimental manipulations developed here can easily be applied to the human brain to test how object identity and configuration uniquely contribute to scene representation in different human visual regions. Such a research endeavor would allow us to document similarities as well as potential differences between the CNNs and the human brain in visual scene processing, and further our understanding of scene representation in both the human brain and CNNs.

## Acknowledgments

We thank Xiaoyu Zhang for creating some of the scene images used here.

## Author Contributions

**Conceptualization:** Kevin Tang, Matthew Chin, Yaoda Xu.

**Data curation:** Kevin Tang, Matthew Chin.

**Formal analysis:** Kevin Tang, Matthew Chin.

**Funding acquisition:** Marvin Chun, Yaoda Xu.

**Investigation:** Kevin Tang, Matthew Chin, Yaoda Xu.

**Methodology:** Kevin Tang, Matthew Chin, Yaoda Xu.

**Project administration:** Yaoda Xu.

**Resources:** Marvin Chun.

**Software:** Kevin Tang, Matthew Chin.

**Supervision:** Yaoda Xu.

**Visualization:** Kevin Tang, Matthew Chin.

**Writing – original draft:** Kevin Tang, Matthew Chin, Yaoda Xu.

**Writing – review & editing:** Kevin Tang, Matthew Chin, Marvin Chun, Yaoda Xu.

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
