## [Decision Letter · Decision Letter 0]

15 Mar 2022

PONE-D-22-01832The contribution of object identity and configuration to scene representation in convolutional neural networksPLOS ONE

Dear Dr. Xu,

Thank you for submitting your manuscript to PLOS ONE. After careful consideration, we feel that it has merit but does not fully meet PLOS ONE’s publication criteria as it currently stands. Therefore, we invite you to submit a revised version of the manuscript that addresses the points raised during the review process. We encourage you to respond to the comments raised by the reviewers, while keeping in mind the PLOS ONE publication criteria. Also, please ensure your paper conforms to the PLOS ONE Data Sharing policy.

We look forward to receiving your revised manuscript.

Kind regards,

Kendrick Kay

Academic Editor

PLOS ONE

Journal Requirements:

(Y.X. is supported by US National Institute of Health (NIH) grant 1R01EY030854.

https://www.nih.gov/

Include this sentence at the end of your statement: The funders had no role in study design, data collection and analysis, decision to publish, or preparation of the manuscript.)

Reviewers' comments:

Reviewer's Responses to Questions

**Comments to the Author**

1. Is the manuscript technically sound, and do the data support the conclusions?

Reviewer #1: Partly

Reviewer #2: Yes

Reviewer #3: Yes

2. Has the statistical analysis been performed appropriately and rigorously? 

Reviewer #1: Yes

Reviewer #2: Yes

Reviewer #3: Yes

3. Have the authors made all data underlying the findings in their manuscript fully available?

Reviewer #1: No

Reviewer #2: Yes

Reviewer #3: No

4. Is the manuscript presented in an intelligible fashion and written in standard English?

Reviewer #1: Yes

Reviewer #2: Yes

Reviewer #3: Yes

5. Review Comments to the Author

Reviewer #1: The manuscript presents a thorough analysis of how different convolutional neural network (CNN) architectures respond to changes in object identity and configuration within a scene. The study appears methodologically sound. The choice of CNNs and variation in the scenes are mostly well-motivated and clearly described. What is unclear in the manuscript is the criteria for the selection of the objects within a scene and the relevance of spatial configuration of objects for scene processing at this level of analysis.

- Previous studies on cortical scene processing emphasize the role of spatial layout of the scene (e.g., open vs. close, navigability of the scene), but what is the relevance of the exact configuration of the objects? A change in spatial configuration of objects likely has a significant effect for a familiar environment, but what is the role for a feedforward scene vision model?

- Could the results also be interpreted as increased position-tolerant object processing across the layers? That is, the first layers are sensitive to configurational and even more to spatial shifts as they are sensitive to changes in low-level visual features whereas the end of the processing stream is more tolerant to such changes?

- More details would be needed about the selection of the objects. Also, the manuscript contains no details about how the scenes were modeled (software? automatic or hand-picked selection of objects?). Is the full set of scene images to be included with the final publication?

- How was the classification accuracy of the scenes defined? More details on what were considered as correct vs. incorrect classifications would be needed.

- The difference between classification accuracy and Euclidean distance results is intriguing. Why would the removal of the background have such drastic effect on classification accuracy but not on the other measures? Can you think this influencing how CNNs are compared with brain data?

Reviewer #2: In this study, the authors present computer generated images of 3D objects arranged in various scene configurations to assess the sensitivity of CNNs to changes in object configuration and object identity. Four different CNN architectures are assessed, and the CNNs were either pre-trained on a task of object classification (ImageNet) or scene classification (Places365). In general, the authors find greater sensitivity to object configuration than to object identity in the early layers of CNNs. This finding is to be expected given that changes in configuration likely lead to larger changes low-level image similarity, as some objects move to new previously unoccupied positions (from inspection of Fig 1). In the final layers for most CNNs, there appears to be greater sensitivity to object identity than to object configuration with the exception of DenseNet. It is claimed that there is a significant interaction effect between background type and layer but from inspection, it seems that the pattern of results are almost identical for no background and fixed background conditions. The authors should perform another analysis just comparing these two conditions because these two are visually the most comparable; the variable background condition introduces unintended shifts in viewpoint that make the interpretation of those results more problematic. From a glance, it appears that object background has little influence, and there does not seem to be any positive evidence to suggest the encoding of object-background relationships.

The authors report very similar results for the object-trained and scene-trained CNNs. While this is in some sense a negative finding, it is a somewhat surprising one. While the analyses and results presented in this paper do not provide a clear explanation for why the results turn out as they do, this finding is of scientific interest and provides potentially relevant information for future studies that wish to compare CNNs trained on objects vs. scenes. It raises the question as to what types of implicit knowledge / representations are these CNNs acquiring through their training, and how similar or different are these when the stimuli and tasks vary?

The comparison of the object- and scene-trained CNNs with untrained CNNs provides positive evidence that sensitivity to object configuration and identity improves with training, and that the profiles observed across layers has some meaningful pattern to it, even though it is difficult to make strong conclusions about the observed pattern. The translation control analysis adds to this interpretation, as translation of a set of objects leads to stronger changes in the responses in the early layers but smaller changes in the near final layers.

In the discussion section, it would be helpful to provide more discussion about sensitivity to object configuration. Does this necessarily represent a genuine sensitivity to configuration or simply some form of implicit representation that is altered by shifts or relative shifts in object position? How might future studies further address or test this issue? The fact that the scene-trained CNNs performed very similarly to object-trained CNNs may suggest that both types of CNNs simply encode objects, with the Place365 CNNs learning multiple objects that are predictive of particular scene categories without acquiring much sensitivity to relative object locations.

Overall, this study reports some interesting patterns and trends across layers regarding sensitivity to object configuration and object identity across layers for object- and scene-trained CNNs. While this study represents an initial foray into these questions, the information it provides will be useful for future investigations, and there will most certainly be many follow-up studies on these types of research questions. The analyses are clearly presented, generally well motivated, with some specific concerns noted (see below). The manuscript is clearly written and accessible. With further revision, I believe that this manuscript could be suitable for publication in PLOS One.

Specific comments

It is not clear whether the data shown in Figure 3 is drawn from object-trained CNNs, scene-trained CNNs, or both.

The use of the term kernel responses is not quite accurate when describing the fully connected layers, also kernels or filters do not respond (they are outputs), recommend rewording.... perhaps unit responses?

The use of z-normalized responses could potentially strongly affect the pattern of results, in particular the pattern of results observed across layers. It would be appropriate to perform the same set of analyses without normalization. Are the findings generally the same or quite different? Would be good to include either as supplementary figures or at least in the response letter to address these concerns.

p. 16 - describing the trends across layers using correlation seems inappropriate. For example, if a very small change occurred across layers but this effect were highly consistent across CNN architectures, this would lead to a very strong positive correlation. Calculating the difference between last and first layers or reporting the slope of change would provide a more interpretable measure.

"All untrained CNNs exhibit a monotonic decline in sensitivity to both types of changes as evidenced by strongly significant negative correlations of both Euclidean distance measures to the rank order of the sampled layer." - this wording is confusing

In the manuscript, it is not clear how the data will be publicly shared.

Reviewer #3: This study reports a set of ‘in silico’ experiments in convolutional neural networks (CNNs), investigating how information about identity and configuration of objects embedded in computer-generated scene images is reflected in the internal activations of CNNs trained for object and scene recognition. The motivation to investigate these two dimensions (identity and configuration) separately and contrast them against one another is that both are thought to contribute to human scene processing, but that their relative importance is unclear. The motivation to study this process in CNNs is that previous work shows that internal activations in CNNs predict responses in scene-selective brain regions in humans. However, the ‘computational algorithms’ involved in scene processing in both CNNs and brains remain unclear, and the authors suggest that investigating how CNNs process different aspects of a scene may shed light on these algorithms.

I reviewed this paper before, for a different journal. I was happy to see that relative to that previous submission, several comments were taken on board here. For example, the motivation for the study now comes across more clearly in the Introduction, the Results section is much more streamlined and readable and now includes important information about CNN accuracy. I see no major concerns with how the experiments were conducted or the statistical analyses and their outcomes. However still some issues remain, related to the logic behind the study, and if it indeed informs us about computational algorithms.

Major comments:

1. Inevitability of results in light of CNN architecture:

A major concern raised in the previous reviews was whether the results could be trivially related to the built-in architecture of the CNNs, which have a form of retinotopy (processing adjacent regions of visual space with adjacent kernels), whereby lower layers have higher spatial resolution compared to higher layers which process increasingly spatially pooled activations. Therefore, it doesn't seem very surprising that conﬁguration information dominates in the earlier layers, and object information in later, fully connected layers that are more closely related to the task objective of labeling the image. I was surprised to see that this issue is not discussed in the current manuscript. I would very much like the authors to discuss their results in light of the architectural design of CNNs, i.e. to provide more theoretical embedding into how they expect the CNN to process these images, given their convolutional and pooling operations. Especially the comparison with untrained CNNs that is included in the current manuscript may speak to this issue, because untrained CNNs share the same architectural features with trained CNNs, but critically the training is lacking.

2. Low accuracy of CNNs on these images:

As mentioned I think it’s good the CNN accuracy is reported and its relation with the comparison of interest (configuration vs. object coding) investigated, but they also raise some questions:

* In the Methods: why only look at the scene-trained CNNs for this? How about the object trained CNNs? Do they classify the objects in the scenes correctly?

* Methods, Page 7: ‘manually assessed whether any of these labels represented a reasonable classification’ – this sounds rather vague, can you give some concrete examples of what you considered reasonable in this context?

* Results: these are mainly focused on whether the classification accuracy impacts the relative representation of object vs. configuration information in the CNN (e.g. Page 13: ‘how a CNN represents object identity and configuration in a scene is not strongly affected by classification accuracy’). But I think the issue here is not necessarily how the accuracy impacts the representation, but more whether it’s meaningful and interesting to look at a CNN if it doesn’t really process these images correctly by any means? 50% accurate for top-5 is not exactly great performance. Why are CNNs then a relevant tool to study computational algorithms of scene processing, if they don’t perform well on the task?

3. Equivalence of CNNs and brain region activations/representations

In a couple of places, some equivalences and comparisons between human/brain/fMRI and CNNs are made that I am uncomfortable with. In my opinion, such equivalences rely on assumptions that are not (yet) empirical grounded - not it this paper (which does not contain any human brain measures or behavior) nor previous literature. For example:

* Page 19: “Activate a scene-like representation in CNNs”; this statement assumes that CNNs have ‘inner representations’ that are scene-like, corresponding to our own introspective notion of seeing scenes as wholes (rather than collections of objects, for example); I’m not sure what such a representation would look like, all CNNs do is find relevant features in images to correctly match it with a prescribed label, whether those features are scene-like or not seems is not (and perhaps cannot be) clearly defined.

* Page 19: “Given that both scene-trained and object-trained CNNs exhibited similar responses in the present study, it remains possible that scene processing in CNNs could be more closely related to the LOC than scene processing in the human brain”; this statement assumes that a scene-trained CNN ‘should be’ more like a scene-processing brain region and an object-trained CNN more like a LOC region. As far as I know it is as of yet unclear whether we can think of these regions as being more or less CNN like (many CNNs seem to have some correlation with all ventral region activity), and to me this prediction reads like an unfounded form of reverse inference from CNNs to brains (I also don’t quite follow why it should go in the direction of LOC: couldn’t it equally well suggest that object processing in CCNs is more related to how PPA processes objects?)

Minor comments:

Abstract:

The conclusion is not really a summary of the finding, but rather a description of the study.

Intro:

* Page 3: Not only Groen et al., showed correspondence between CNN features and fMRI responses in scene regions in the brain, but also Bonner & Epstein 2017 (PNAS), Dwivedi 2020 (JOCN) King et al. 2019, (NeuroImage), to name a few.

* Page 3: ‘How these two types of information are weighted during scene processing has not been systematically examined in either the human brain or CNNs’. That’s quite a statement to make – especially for CNNs, there are so many publications on deep learning across many conferences and journals… So maybe better to say ‘to our knowledge, this has not been examined….’?

* Page 4: How is similarity between brain and CNN ‘utilized’ in this study? Do you perhaps mean ‘motivated by’?

Methods:

* Page 5: the motivation for CNN selection still very brief, a bit more background (e.g. what are residual blocks in ResNets and why are they ‘novel’?) would be helpful. A phrase like ‘were chosen for their convolution depth’ is not really informative; why is convolution depth interesting/important to consider when investigating scene processing?

* Page 5: ‘inside the convolution’ – awkward phrasing, ‘in the hierarchy’ would be more appropriate, I think.

* Page 7: ‘reversal of magnitude’ – magnitude of what? Unit activations?

Results:

* Page 11: ‘All statistical tests performed used image set as dependent variable’ – this seems inaccurate to me, the dependent variable is Euclidean distance. Do you mean that the image sets are the individual data points going into the statistical test (and that the error measures reflect the error across image sets)?

* Page 11: ‘near significant’ is not a thing… you can talk about a trend perhaps.

* What are the units in Table 2? Percent (0-100), or a score between 0 and 1?

Discussion:

* Page 18: “But also can shed significant insights on the likely computational algorithms used by the human brain during scene processing”. I would be careful claiming this, because a) human scene processing is not investigated in this study and b) we don’t know if insights acquired by doing so would indeed be ‘significant’.

* Page 19: “Removing spatial background had only minor effects on the sensitivity of a CNN on object or configuration changes” – but it had a big effect on accuracy, isn’t that more relevant when considering activity in PPA?

6. PLOS authors have the option to publish the peer review history of their article (what does this mean?). If published, this will include your full peer review and any attached files.

Reviewer #1: No

Reviewer #2: No

Reviewer #3: No

---

## [Author Response · Author response to Decision Letter 0]

29 Apr 2022

Please see the attached response letter.

---

## [Decision Letter · Decision Letter 1]

15 Jun 2022

The contribution of object identity and configuration to scene representation in convolutional neural networks

PONE-D-22-01832R1

Dear Dr. Xu,

We’re pleased to inform you that your manuscript has been judged scientifically suitable for publication and will be formally accepted for publication once it meets all outstanding technical requirements.

Kind regards,

Kendrick Kay

Academic Editor

PLOS ONE

Additional Editor Comments (optional):

Dear authors,

Congratulations: your manuscript is suitable for publication in PLOS One. Thank you for your submission.

Kendrick Kay

PLOS ONE Academic Editor

Reviewers' comments:

Reviewer's Responses to Questions

**Comments to the Author**

1. If the authors have adequately addressed your comments raised in a previous round of review and you feel that this manuscript is now acceptable for publication, you may indicate that here to bypass the “Comments to the Author” section, enter your conflict of interest statement in the “Confidential to Editor” section, and submit your "Accept" recommendation.

Reviewer #1: All comments have been addressed

Reviewer #2: All comments have been addressed

Reviewer #3: (No Response)

2. Is the manuscript technically sound, and do the data support the conclusions?

Reviewer #1: Yes

Reviewer #2: Yes

Reviewer #3: Partly

3. Has the statistical analysis been performed appropriately and rigorously? 

Reviewer #1: Yes

Reviewer #2: Yes

Reviewer #3: Yes

4. Have the authors made all data underlying the findings in their manuscript fully available?

Reviewer #1: Yes

Reviewer #2: Yes

Reviewer #3: Yes

5. Is the manuscript presented in an intelligible fashion and written in standard English?

Reviewer #1: Yes

Reviewer #2: Yes

Reviewer #3: Yes

6. Review Comments to the Author

Reviewer #1: (No Response)

Reviewer #2: The authors have been responsive to the reviews and the revised discussion section expands on a some of the bigger questions that were raised. As noted before, the control analyses involving untrained CNNs and also the translation control manipulation provide evidence of greater sensitivity (likely implicit but still to be determined) to object configuration in the later layers of the trained CNNs. Overall, this is a very nice study and I have no further concerns.

Reviewer #3: I don’t have any remaining major comments. Below are just two minor suggestions and one general reflection on the paper.

Line 175: ‘have also been shown to classify different room scenes with high accuracy’ – a citation would be appropriate here.

The Discussion section on Line 858 (on the comparison between object- and scene-trained CNNs): here it could be helpful to discuss some of the work from Bolei Zhou (together with Aude Oliva and Antonio Torralba) who used a technique called ‘network dissection’ (see Bau et al., 2020, PNAS) to suggest that place-trained CNNs develop whole-object detectors (e.g. https://arxiv.org/pdf/1412.6856.pdf).

Finally, more of a general comment that the authors don’t need to act upon, but that I still feel I should mention. I remain somewhat puzzled about the way the authors think about CNNs. The paper is motivated by the idea that CNNs can inform us about Marr’s algorithmic level of visual processing. From some of the new text - e.g. on lines 862 and 868 - I get the impression the authors conceive of CNNs as systems that over the course of learning ‘develop’ new processing algorithms in order to construct something like mental ‘representations’ (e.g. of a configuration of objects). As far as I understand (but I might be wrong), all CNNs do is learn to map combinations of pixels to an output label through some sequence of non-linear filtering operations. In other words, I don’t think that over the course of training, they learn what the authors refer to as a higher-order ‘processing algorithm’– the computational algorithm is always the same, namely filtering, ReLU, pooling, weights, etc. The learning rule is also always the same, back-propagating the loss on the label accuracy. Personally, while I think it’s interesting to use CNNs see what information in the image seems to be useful for categorization or for explaining brain responses, I think we should be careful to place them on too equal footing to biological systems that are a lot more ‘black-box’ in the sense that we don’t know the computations or the learning rules. Again, I don’t think the authors need to address this in further Discussion- perhaps we just have different interpretation of what ‘algorithmic’ means - but I just wanted to highlight this as something to consider as the use of CNNs as models of brain computation becomes more widespread.

7. PLOS authors have the option to publish the peer review history of their article (what does this mean?). If published, this will include your full peer review and any attached files.

Reviewer #1: No

Reviewer #2: No

Reviewer #3: No

---

## [Editor Report · Acceptance letter]

20 Jun 2022

PONE-D-22-01832R1 

The contribution of object identity and configuration to scene representation in convolutional neural networks 

Dear Dr. Xu:

I'm pleased to inform you that your manuscript has been deemed suitable for publication in PLOS ONE. Congratulations! Your manuscript is now with our production department. 

Kind regards, 

on behalf of

Dr. Kendrick Kay 

Academic Editor

PLOS ONE